# Effect of Culture Medium Composition on In Vitro Regeneration, Acclimatization, and Production Cost of *Dendrobium phalaenopsis* Sa-Nook ‘Thailand Black’ Plants

**DOI:** 10.3390/plants15010088

**Published:** 2025-12-27

**Authors:** José Trinidad Zavala-Hernández, Alejandrina Robledo-Paz, Víctor A. González-Hernández, María Alejandra Gutiérrez-Espinosa, Martín Mata-Rosas

**Affiliations:** 1Colegio de Postgraduados, Postgrado en Recursos Genéticos y Productividad, Campus Montecillo, Texcoco 56264, Estado de Mexico, Mexico; trinozavala@gmail.com (J.T.Z.-H.);; 2Red de Manejo Biotecnológico de Recursos, Instituto de Ecología AC, Xalapa 91073, Veracruz, Mexico; martin.mata@inecol.mx

**Keywords:** orchid, micropropagation, cytokinins, auxins, ex vitro growth, cost

## Abstract

The *Dendrobium* genus originates from tropical and subtropical regions of Asia, extending to northern Australia. Species and hybrids of this genus, including *Dendrobium phalaenopsis* Sa-Nook ‘Thailand Black’, are the second most popular cultivars worldwide. To meet this demand, it is necessary to implement techniques such as micropropagation, which allows the mass production of plants. The objective of this study was to evaluate the effect of plant growth regulators (NAA, BA, and TDZ) on the in vitro regeneration of *D. phalaenopsis* Sa-Nook ‘Thailand Black’ plants, their acclimatization, and production costs. Rootless shoots of 1–1.5 cm in height were grown on 50% MS medium supplemented with BA (0, 2.22, 4.44, or 8.88 µM), TDZ (0, 2.27, 4.54, or 9.08 µM), and NAA (0 or 2.69 µM), individually or in combination. After the second phase of in vitro shoot growth, the highest number of shoots per explant (8.5) was observed with 9.08 µM of TDZ. Plants regenerated with this concentration of TDZ showed the highest survival rate (96%) at 90 days of greenhouse cultivation, as well as the formation of new shoots (0.9), and the lowest production cost per plant (0.49 USD).

## 1. Introduction

Orchids are a group of plants consisting of approximately 27,000 species and nearly 1000 genera [1]. Habitat loss, illegal collection from wild populations, and high commercial, medicinal, and edible demand have led to the Orchidaceae family being included in Appendix II of the Convention on International Trade in Endangered Species of Wild Fauna and Flora (CITES) [2]. Within the Orchidaceae family, *Dendrobium* is the second largest genus in terms of number of species (1600 epiphytic, lithophytic, and terrestrial species), and hybrids [3,4]. This species is common in tropical and subtropical regions of Asia and northern Australia [5,6].

Globally, in the legal orchid trade, *Dendrobium* and all its hybrids are of great floricultural importance, representing 85% of the cut flower orchid trade [7,8,9]. *Dendrobium phalaenopsis* is native to Indonesia and is preferred over *Phalaenopsis* due to its fast growth, abundant flowers, wide range of colors, sizes, and shapes, as well as its year-round availability and long flowering life. This species is mainly cultivated in the Southeast Asian region, such as Thailand, Singapore, and Malaysia [10,11]. Hybrids derived from the Sa-Nook line of *D. phalaenopsis* were improved to accelerate growth and optimize their flowering capacity. The ‘Thailand Black’ variant is characterized by its large, dark violet flowers that can reach 6–6.5 cm in size [12].

To address the commercial demand for orchids and reduce the exploitation of wild populations, micropropagation has been crucial for the large-scale cultivation of endangered species. It has also played a significant role in the recovery of wild populations [13]. An important factor in micropropagation is the use of plant growth regulators (PGRs), since PGRs influence the reprogramming of cell development and the differentiation process [6].

There are only a few studies regarding the micropropagation of *Dendrobium phalaenopsis*. In some of these protocols, different organs were used as explants (protocorms, shoot tips, axillary buds, and microshoots), and the effect of PGRs such as kinetin (K), N6-benzyladenine (BA), 1-naphthaleneacetic acid (NAA), and indole-3-butyric acid (IBA) was evaluated [14,15,16,17,18,19]. However, the effect of thidiazuron (TDZ) to induce adventitious shoots in young plants has not been tested, and there are no reports on the effect of TDZ and BA, alone or combined with NAA, in the induction of shoots.

The measure of success of any micropropagation protocol lies not only in achieving high multiplication rates, but also in the fact that the in vitro plantlet survives the acclimatization process in ex vitro environmental conditions [20]. This is justified because in vitro regenerated plants may present abnormalities and suffer anatomical, morphological, and physiological alterations, which have a negative impact on their quality and survival ex vitro [21,22].

Although micropropagation protocols have been reported for species of the genus *Dendrobium*, to the best of our knowledge, only one has been reported for *Dendrobium phalaenopsis* [15]. Furthermore, the effect of PGR on plant regeneration, plant acclimatization behavior, and production costs has not been evaluated. The objective of this research was to evaluate the effect of plant growth regulators (BA, TDZ, and NAA) both individually and in combination, on the in vitro regeneration efficiency of *D. phalaenopsis* Sa-Nook ‘Thailand Black’, as well as the acclimatization of the regenerated plants and their production cost.

## 2. Results

### 2.1. Shoot Induction

After 15 days of culture, the development of small green spherical bodies similar to protocorms (PLB) attached to the base of the explant was observed in the explants of all treatments, being greater in those explants cultivated in the presence of any concentration of TDZ combined with NAA. At the end of the shoot induction phase, the PLB developed and began to elongate, resulting in the formation of shoots.

After transforming the values of the evaluated variables using the natural logarithm, the assumptions of normality and homogeneity of variances were met; therefore, differences among treatments were analyzed using analysis of variance (ANOVA), followed by Tukey’s multiple comparison test. Data analysis revealed significant differences among treatments in two of the three evaluated variables (explants forming shoots and number of shoots per explant) (Table 1). The percentage of explants forming shoots when cultured in the presence of 4.44 and 8.88 µM BA, 2.69 µM NAA + 8.88 µM BA, as well as 2.69 µM NAA combined with any of the tested TDZ concentrations, was significantly higher (91.4–100%) than in the remaining treatments. The highest number of shoots per explant was observed with 9.08 µM TDZ (2.5), with no significant differences compared to treatments containing 4.44 and 8.88 µM BA, 2.69 µM NAA combined with 2.22 or 8.88 µM BA, 2.69 µM NAA combined with any TDZ concentration, and the control (Table 1).

### 2.2. Shoot Growth

#### 2.2.1. First Phase

After the first phase of shoot growth, the Shapiro–Wilk and Bartlett goodness-of-fit tests, as well as the transformed tests (natural logarithm and square root), revealed that none of the variables evaluated in this phase met the assumptions of normality and homogeneity of variances. Therefore, the differences between treatments were estimated using the Kruskal–Wallis test and the Wilcoxon rank test. Significant differences were detected between treatments in three of the five variables evaluated (shoots per explant, shoot length, number of roots per shoot) (Table 2).

One hundred percent of the explants from all treatments formed shoots. The highest number of shoots per explant (5.2) was obtained with 9.08 µM TDZ, which was significantly different from the rest of the treatments (Table 2, Figure 1B); while the lowest value for both variables was observed with 2.69 µM NAA + 4.44 µM BA (Table 2, Figure 1C). Explants grown in the presence of TDZ alone or in combination with NAA formed more shoots than when BA or NAA (Figure 1D–F) were used independently or in combination or when no growth regulators were used (control, Figure 1A).

Shoots that differentiated on media with BA were longer than those regenerated in the presence of NAA or TDZ, either independently or in combination. The length of shoots regenerated in a PGR-free medium, or in the presence of any of the tested concentrations of BA or NAA alone or in combination, was statistically greater than that of shoots obtained with TDZ alone (Table 2). The highest number of roots per shoot was recorded with any of the BA concentrations tested individually or in combination with NAA, without these values being significantly different from each other, and the control. Shoots obtained with TDZ formed the lowest number of roots, except with the combination of 2.27 μM TDZ + 2.69 μM NAA (Table 2).

#### 2.2.2. Second Phase

After the assumptions of normality and homogeneity of variances were validated, differences among treatments were analyzed using analysis of variance (ANOVA), followed by Tukey’s multiple comparison test. The analysis revealed significant differences in three of the five evaluated variables (shoots per explant, shoot length, and number of roots per shoot) (Table 3). At the end of the second growth phase, the highest number of shoots per explant was observed with 9.08 µM TDZ (8.5), which was significantly different from the remaining treatments. In contrast, the highest shoot length values were recorded in the presence of 4.54 µM TDZ and in the control. Shoots regenerated with 2.69 µM NAA + 2.22 µM BA (7.4), 2.27 µM TDZ (6.7), and the control (7.1) exhibited a significantly higher number of roots compared to the other treatments (Table 3).

### 2.3. Acclimatization Under Greenhouse Conditions

The Shapiro–Wilk normality goodness-of-fit test revealed that four of the seven variables evaluated did not meet the assumptions. However, when transformed to natural logarithm (survival and formation of new shoots) and square root (number and length of plant leaves), the assumptions of normality and homogeneity of variances were validated; therefore, the differences between treatments were estimated by ANOVA and Tukey’s multiple comparison tests of means. Plants from each in vitro regeneration treatment were acclimatized. After 90 days of acclimatization, significant differences were found between treatments in all the variables measured: survival, formation of new shoots, number of roots, number of leaves, root length, leaf length, and plant length.

At the end of acclimatization (90 d), it was found that survival was higher (96%) in those plants regenerated in the presence of TDZ (9.08 μM), with no statistical differences compared to those regenerated with NAA or the combination of BA and NAA (Table 4, Figure 2B). In addition to observing higher survival, these plants also formed the largest number of new shoots (0.91) under greenhouse conditions, which represents values of 84 and 203% higher, respectively, than those recorded for these same variables in the control plants (Table 4, Figure 2A).

The multiple comparison of means with the Tukey test allowed for detecting differences in the morphological variables. It is noteworthy that, although the plants from all in vitro treatments managed to increase the number of roots at the end of acclimatization, the plants that were regenerated with 2.69 μM of NAA showed a rate of change (RC) of 60.9% concerning the number of roots they had at the beginning of acclimatization (Table 4, Figure 2C). Differences were observed in the number of leaves after acclimatization, where plants regenerated with 2.69 μM NAA + 2.22 μM BA achieved the highest number of leaves, representing a RC of 4.4% compared to the initial number of leaves (Table 4, Figure 2D).

Plants regenerated in the absence of PGRs showed the highest positive RC values for root length (0.8%) (Table 5, Figure 2A). For leaf length, the highest positive RC value was obtained with 2.27 µM TDZ (1.6%) (Table 5, Figure 2F). As observed for roots, most leaves developed in vitro died, and those that were developing at the beginning of acclimatization were the ones that adapted to the new conditions; although they continued to grow, they did not exceed the size of the leaves that died.

At the end of acclimatization, all plants increased their shoot length, with significant differences among treatments. Plants showing the highest RC values for shoot length were those obtained with 4.44 µM BA (23.9%) (Table 5, Figure 2E).

### 2.4. The Production Costs

The production costs per plant regenerated in vitro until acclimatization showed significant differences between treatments after transforming the amounts with the square root and validating the assumptions of normality and homogeneity of variance. Plants regenerated using various combinations and concentrations of growth regulators had a lower cost, or no significant differences in cost, compared to plants cultivated in media without plant growth regulators (control), which cost 2.22 USD. Only plants regenerated with 4.44 μM BA had production costs exceeding those of control plants by 217% (4.82 USD). In contrast, the plants grown with 9.08 μM TDZ exhibited the lowest production cost at 0.49 USD (Table 6).

The number of plants that were regenerated and acclimatized using one liter of induction medium (NPL) was the key factor influencing the variation in production costs per plant across different treatments (Table 6). Additionally, the NPL was directly linked to the number of shoots produced per explant as well as the survival rate of plants once they were removed from the controlled environment.

## 3. Discussion

Multiple shoot regeneration and proliferation are closely related to the type and concentration of cytokinins and auxins applied [2,23,24]. In this study, it was found that during the induction phase (30 days of culture), the number of regenerated shoots depended on the type of PGRs and their concentration in the culture medium; the highest number of regenerated shoots per explant was observed with 9.08 µM TDZ (2.5). In this regard, Asghar et al. (2011) in *D. nobile* var. ‘Emma White’ reported the formation of the highest number of shoots (4.3) when cultured with 8.88 μM of BA [14]. Likewise, Pathak et al. (2022) in *D. chryseum* found the highest number of shoots (5.8) in explants cultured with 2.22 μM BA + 2.69 μM NAA for 16 weeks [17]. At the same time, Liu et al. (2023) obtained the highest number of shoots (6.1) from *D. moniliforme* with 4.44 μM BA + 5.37 μM NAA after 14 weeks of culture [18]. Subrahmanyeswari et al. (2022) observed the highest number of shoots per explant (12.3) when *D.* var. ‘Yuki White’ explants were cultured in the presence of 2.22 μM BA + 0.54 μM NAA + 217.2 μM adenine sulfate for eight weeks [25].

Unlike what was observed in other *Dendrobium* species, it was possible to observe that TDZ was more efficient in the induction of shoots in *D. phalaenopsis* Sa-Nook ‘Thailand Black’, even at lower concentrations than those of BA and NAA. The high efficacy of TDZ applied at low doses can be explained by its ability to resist degradation induced by the enzyme cytokinin oxidase [26], which allowed TDZ levels to remain stable for longer [27,28]. It is clear that the morphogenetic responses of the *Dendrobium* genus to TDZ can vary between the same species, hybrids, or cultivars, presumably due to genetic differences, endogenous hormone content, and the type of explant.

Just as the number of shoots obtained during the induction phase depended on the type and concentration of PGRs, the number of shoots, shoot length, and number of roots during the first and second shoot growth phases were affected by the type and concentration of the PGR tested during induction, even though these regulators were no longer present in the shoot growth medium.

Regarding the root, it was found that TDZ inhibited its development during in vitro culture (first phase of shoot growth); this inhibition increased as the concentration of TDZ in the culture medium increased. This trend was maintained when TDZ was applied in combination with NAA; in contrast, the plants in the BA treatments developed more roots than the control. The shoots that formed a few roots during the first phase of shoot growth developed an adequate root system when they were cultured in the medium containing activated charcoal during the second phase of shoot growth. The inhibition of the root system in shoots regenerated in the presence of TDZ could be because TDZ promotes the endogenous production of ethylene, a hormone that inhibits auxin transport and, therefore, the formation of the root system [29]. The inhibition of the root system induced by TDZ was also observed in *Dendrobium wilsonii* Rolf [30].

On the other hand, shoot size was favored by the presence of BA alone or combined with NAA. Shoots regenerated with TDZ developed a root system in a similar quantity to the rest of the shoots when they were cultured on culture medium containing activated charcoal in the second phase of shoot growth. The activated charcoal was utilized to absorb the phenols and ethylene released by the shoots, leading to improved development of both the aerial and root systems. Díaz et al. (2010) indicate that for orchid in vitro plantlets to ensure their survival after being transferred to ex vitro conditions, it is essential that they develop a sufficient number of leaves and a strong root system during the in vitro growth phase [31].

The conditions that predominate in vitro culture (aseptic environment, controlled temperature, high relative humidity, high nutrient availability, low light intensity, and low CO_2_ concentration) can generate morphological, anatomical, and physiological changes in the regenerated plants, such as: low photosynthetic rates, poor functioning of the stomatal apparatus, and reduction of the epicuticular wax in their leaves, which causes excessive evapotranspiration when transplanting them to ex vitro conditions [20,22].

Plants regenerated with any of the TDZ concentrations tested were slightly smaller at the beginning of acclimatization and had shorter and fewer roots than those in the other treatments; however, the number of leaves was similar in all plants regardless of the medium in which they were regenerated. Since plants regenerated in the presence of TDZ (9.08 µM) had a higher survival rate after 90 d of remaining in greenhouse conditions than those cultured with BA and NAA, it can be inferred that TDZ contributed to their better in vitro development, which allowed them to adapt more quickly to the new growth conditions. The aforementioned was reflected in the formation of new shoots, which could be an indication that the plants adapted well to the greenhouse conditions and had begun a new stage of growth.

Since the roots and leaves of plants grown in vitro are not adapted to greenhouse conditions, the acclimatization process involves the formation and development of new leaves and roots, on which the success of their growth in these new conditions depends [22]. After 90 days of growing the plants in a greenhouse, the development of new and small roots was observed, which caused the average length to decrease. On the contrary, the number of leaves and their length showed a general loss since the plants lost the leaves that had previously developed in in vitro conditions. In this regard, Paul et al. (2017) observed that during the acclimatization of *Dendrobium fimbriatum* plants, the older leaves died with the appearance of new leaves that were adapted to the ex vitro conditions [32].

This is the first protocol in the *Dendrobium phalaenopsis* species where the effect of PGR on micropropagation is reported. In the micropropagation protocols reported in different species of the *Dendrobium* genus, the effect of in vitro culture conditions (growth regulators) used to regenerate plants, on the acclimatization process, has not been evaluated, perhaps because it is thought that this effect disappears over time. In this way, the plants are transplanted to a substrate without analyzing whether the culture medium used to regenerate them has any effect on their characteristics, survival, and growth during acclimatization. In the present research, traceability was given to the plants that came from each treatment when they were transferred to the greenhouse. The results indicated that the PGRs applied during the induction phase affect the characteristics of the regenerated plants and their behavior during acclimatization.

Micropropagation is a technique that enables the multiplication of economically and ecologically important orchids, both wild and cultivated [23,25,33,34,35,36]. This technique has made it possible to satisfy the global demand for these and other ornamental plant species. However, the production cost of regenerated in vitro plantlets until acclimatization has rarely been reported [37,38,39,40,41,42,43]. Knowing the costs of regenerated plants is an important aspect of the production chain. This knowledge helps to determine whether the developed protocol is efficient enough to compete with conventional propagation methods of the species of interest, or if it needs to be modified in order to reduce costs. The results obtained from *D. phalaenopsis* Sa-Nook ‘Thailand Black’ indicate that the factors that most significantly influence the production cost of regenerated plants are the number of shoots per explant and the survival rate in ex vitro conditions. This finding is consistent with that reported in *Theobroma cacao*, *Gynerium sagitatum*, and *Phalaenopsis* sp. [38,41,42].

In the Mexican market, a *Dendrobium* in vitro plantlet imported from Thailand costs approximately 1.17 USD, which is 138% more than the cost of an acclimatized *D. phalaenopsis* Sa-Nook ‘Thailand Black’ micropropagated plant produced with the protocol reported here (0.49 USD). This indicates that the implementation of this protocol is affordable and would allow for a considerable profit margin. On the other hand, Henao-Ramírez et al. (2022) reported a cost of 0.73 USD for an acclimatized *Theobroma cacao* plant produced in vitro [42]. For *Coffea canephora*, *Capsicum chinense*, and *Gynerium sagitatum*, the approximate costs per plant were 0.19 USD, 0.23 USD, and 0.18 USD, respectively [37,39,41]. In contrast, the cost of *Phalaenopsis sp*. plants regenerated in vitro was 0.30 USD [38].

In the present investigation, a micropropagation protocol was established for *D. phalaenopsis* Sa-Nook ‘Thailand Black’, which consists of culturing young plants (1.0–1.5 cm) without roots, on MS basal medium supplemented with 9.08 μM TDZ for 30 d. The explants are then transferred to a basal medium without PGRs for 90 d (first phase of shoot growth); the individual shoots are subcultured for 120 d in a basal medium with 1 g L^−1^ of activated charcoal (second phase of shoot growth). The plants are initially pre-acclimated for 5 d, followed by acclimatization in *Sphagnum* moss for 90 d inside an acclimatization tunnel within the greenhouse. This protocol guarantees the mass propagation of plants with characteristics that allow their successful adaptation to greenhouse or nursery conditions with the lowest production cost per plant.

## 4. Materials and Methods

### 4.1. Plant Material

*Dendrobium phalaenopsis* Sa-Nook ‘Thailand Black’ plants of 1.0–1.5 cm height with three or four leaves, without roots, were used. These plants were obtained through seed germination on Knudson C Orchid [44] culture medium at 50% concentration (Caisson Lab, Smithfield, Utah, USA), supplemented with 2% sucrose, 0.1% activated charcoal (Meyer, Mexico City, Mexico), and 0.7% Type I agar (Caisson Lab, Smithfield, UT, USA).

### 4.2. Culture Medium

The basal culture medium consisted of Murashige and Skoog (MS) [45] salts (Caisson Lab, Smithfield, UT, USA) 50%, supplemented with 3.0% sucrose and 0.8% Type I agar (Caisson Lab, Smithfield, UT, USA). The pH of the medium was adjusted to 5.7 before adding the agar, and it was sterilized in an autoclave (Model FE-406, Felisa^®^, Zapopan, Jalisco, Mexico) at 120 °C (20 PSI) for 20 min.

### 4.3. Induction of Adventitious Shoots

The explants were placed in 125 mL polypropylene containers containing 20 mL of basal medium without PGRs (control) or supplemented with different concentrations of BA (0, 2.22, 4.44, 8.88 µM), TDZ (0, 2.27, 4.54, 9.08 µM), and NAA (0, 2.69 µM) (Sigma-Aldrich, St. Louis, MO, USA) individually or in combination. Under these conditions, the explants remained for 30 days for shoot induction. After 30 d of culture, the percentage of survival, the percentage of explants that formed shoots, and the number of shoots per explant were evaluated.

### 4.4. Shoot Growth

#### 4.4.1. First Phase

Explants that formed shoots were transferred to 125 mL polypropylene containers containing 20 mL of basal medium, in which they were maintained for 90 d to promote shoot growth.

#### 4.4.2. Second Phase

Shoots were then individualized and transferred to 500 mL polypropylene containers with 70 mL of basal medium (Caisson Lab, Smithfield, UT, USA) supplemented with 1 g L^−1^ activated charcoal (Meyer, Mexico City, Mexico), in which they remained for a further 120 d. After the first and second growth phases, the percentage of survival, the percentage of explants that formed shoots, the number of shoots per explant, shoot length, and the number of roots per shoot were evaluated.

### 4.5. Culture Incubation Conditions

All cultures were kept in a growth chamber at 22 ± 4 °C, with a 16 h photoperiod provided by LED lamps (60 W, Philips^®^, Jiujiang, China), and photosynthetically active radiation of 55 µmol m^2^ s^−1^ (Li-Cor^®^ Quantum Sensor model LI-190R coupled to a Light Sensor Logger, model LI-1500, Lincoln, NE, USA).

### 4.6. Acclimatization

After the second phase of shoot growth (240 d since the in vitro culture was established), the plants were first subjected to a pre-acclimatization period, which consisted of placing the containers with the plants produced in vitro, closed, in an acclimatization tunnel inside the greenhouse for 5 d. Then the plants were removed, washed with running water to remove the agar adhering to the roots, and immersed in a 0.1% fungicide solution (Carbendazim, Helm^®^, State of Mexico, Mexico) for 20 min. The plants were transplanted into plastic trays with 50 cavities of 60 cm^3^ containing *Sphagnum* moss (Chilemoss, Puerto Montt, Chile) as a substrate. The trays were placed inside an acclimatization tunnel in which the ambient humidity was gradually reduced (95 to 65 ± 10%) by progressively opening the tunnel until it was completely open after the seventh week of acclimatization. Every seven days, the plants were watered with running water, and a 0.1% amino acid solution (Lidamino^®^, Lida, Valencia, Spain) was applied, and every 15 days, a 0.025% mixture of vitamins and hormones (SUPERthrive^®^, North Hollywood, CA, USA). At the beginning and at the end of acclimatization (90 days), the following were evaluated: the number and length of leaves, number and length of roots, the number and length of leaves, the length of the plant, the formation of new shoots, and the percentage of plant survival.

### 4.7. Production Cost

The production cost per regenerated and acclimatized plant was estimated using the activity-based costing (ABC) system [46], which considers direct and indirect costs. Direct costs include direct expenses for the acquisition of materials used in in vitro culture and growth under greenhouse conditions, explants, utilities (electricity and water), direct labor, etc. Indirect costs consider indirect materials (such as materials for washing, work area cleanliness, and explant disinfestation), utilities (such as electricity, water, and telephone), indirect labor, depreciation of equipment and infrastructure, and maintenance expenses. The cost per plant was estimated based on the percentage of explants that formed shoots, the number of shoots per explant, the quantity of plants regenerated from one liter of shoot induction medium, and the survival rate of these plants under greenhouse conditions after 90 days.

### 4.8. Experimental Design and Statistical Analysis

A completely randomized experimental design with 14 treatments was used for the shoot induction and shoot growth phases (resulting from the concentrations of NAA, BA, and TDZ individually and combined, plus a control without PGRs), with seven repetitions per treatment, where the experimental unit was a flask with five explants (35 explants per treatment). For the acclimatization phase, the same 14 treatments of the shoot induction phase were evaluated, but with five repetitions and five plants per experimental unit (25 plants per treatment). Finally, the cost per regenerated plant was estimated in each of the 14 treatments tested, with five replicates per treatment.

The statistical analysis was performed in the open-source software SAS OnDemand (https://welcome.oda.sas.com/) [47]. For all variables of the induction, shoot growth, and acclimatization phases, as well as the cost per plant, the assumptions of normality and homogeneity of variances were checked; if they were not met, logarithmic and square root transformations were performed to normalize the data. When it was not possible to normalize the data with the transformations, the nonparametric Kruskal–Wallis test was used to determine significant differences between treatments; the differences between their medians were estimated by means of the Wilcoxon rank test. The values of the variables that met the assumptions of normality and homogeneity of variance were subjected to an analysis of variance (ANOVA) and multiple comparison of means with Tukey (*p* ≤ 0.05).

## 5. Conclusions

The developed protocol allows the successful in vitro propagation of *D. phalaenopsis* Sa-Nook ‘Thailand Black’, using a culture medium supplemented with 9.08 μM TDZ, with which the highest number of plants can be induced. The plants obtained under these culture conditions showed characteristics that allowed them to optimally adapt to ex vitro conditions, which was reflected in their high survival rate and growth during acclimatization, in addition to the lower production cost per plant. This protocol will allow the efficient multiplication of plants to meet the commercial demand for this hybrid.

## Figures and Tables

**Figure 1 plants-15-00088-f001:**
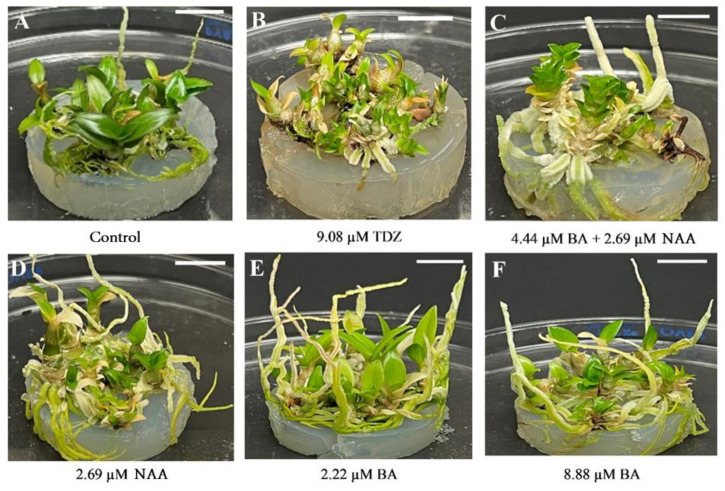
Effect of 1-naphthaleneacetic acid (NAA), N6-benzyladenine (BA), and thidiazuron (TDZ) on in vitro shoot regeneration of *Dendrobium phalaenopsis* Sa-Nook ‘Thailand Black’, after the first phase of shoot growth. (**A**) PGR-free; (**B**) 9.08 µM TDZ; (**C**) 4.44 µM BA + 2.69 NAA; (**D**) 2.69 µM NAA; (**E**) 2.22 µM BA; (**F**) 8.88 µM BA. Bars = 1 cm.

**Figure 2 plants-15-00088-f002:**
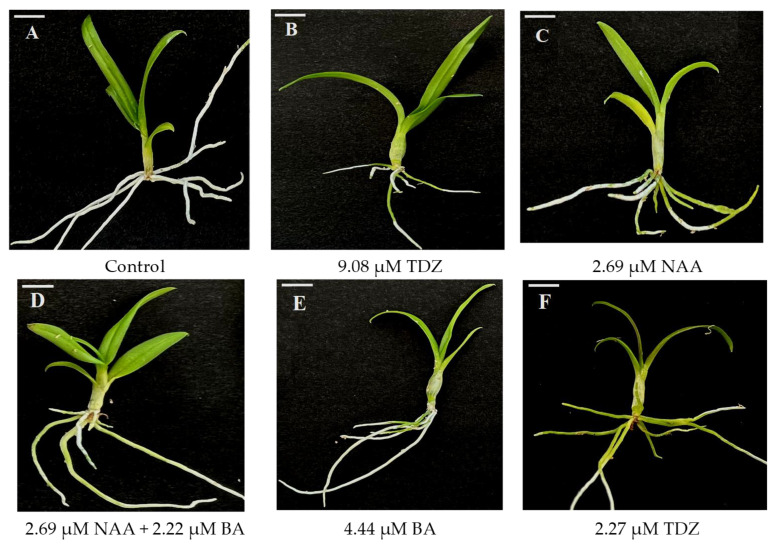
Effect of 1-naphthaleneacetic acid (NAA), N6-benzyladenine (BA), and thidiazuron (TDZ) used in the in vitro propagation stage of *Dendrobium phalaenopsis* Sa-Nook ‘Thailand Black’ on the performance of regenerated plants under greenhouse conditions, after 90 days of cultivation. (**A**) PGR-free; (**B**) 9.08 µM TDZ; (**C**) 2.69 µM NAA; (**D**) 2.69 µM NAA + 2.22 µM BA; (**E**) 4.44 µM BA; (**F**) 2.27 µM TDZ. Bars = 1 cm.

**Table 1 plants-15-00088-t001:** Effect of growth regulators (BA, TDZ, and NAA) on in vitro shoot induction of *Dendrobium phalaenopsis* Sa-Nook ‘Thailand Black’, after 30 days of culture.

NAA	BA	TDZ	SUR	SFE	SE
(µM)			(%)	(%)	
0	0	0	100.00 a	62.86 bc	2.06 abc
0	2.22	0	100.00 a	57.14 c	1.63 c
0	4.44	0	100.00 a	94.29 a	1.99 abc
0	8.88	0	100.00 a	94.29 a	2.09 abc
0	0	2.27	100.00 a	71.43 bc	1.90 bc
0	0	4.54	100.00 a	77.14 bc	1.79 bc
0	0	9.08	100.00 a	74.29 bc	2.55 a
2.69	0	0	100.00 a	80.00 bc	1.87 bc
2.69	2.22	0	100.00 a	74.29 bc	1.99 abc
2.69	4.44	0	100.00 a	85.71 b	1.77 bc
2.69	8.88	0	100.00 a	100.00 a	2.37 ab
2.69	0	2.27	100.00 a	100.00 a	2.34 ab
2.69	0	4.54	100.00 a	91.43 ab	2.08 abc
2.69	0	9.08	100.00 a	91.43 ab	2.00 abc

Means with the same letter in each column are not statistically different (Tukey, *p* < 0.05). NAA: 1-naphthaleneacetic acid, BA: N6-benzyladenine, TDZ: thidiazuron, SUR: survival, SFE: shoot-forming explants, SE: number of shoots per explant.

**Table 2 plants-15-00088-t002:** Effect of BA, TDZ, and NAA on shoot growth of *Dendrobium phalaenopsis* Sa-Nook ‘Thailand Black’ (first phase).

NAA	BA	TDZ	SUR	SFE	SE	SL	NRS
(µM)			(%)	(%)		(mm)	
0	0	0	100.00 a	100.00 a	2.35 cd	15.89 a	6.42 abc
0	2.22	0	100.00 a	100.00 a	2.40 cd	17.88 a	8.25 a
0	4.44	0	100.00 a	100.00 a	2.60 cd	14.95 ab	7.03 ab
0	8.88	0	100.00 a	100.00 a	2.30 cd	15.37 a	8.47 a
0	0	2.27	100.00 a	100.00 a	3.20 bc	9.72 c	4.54 cd
0	0	4.54	100.00 a	100.00 a	3.50 bc	10.21 c	3.88 de
0	0	9.08	100.00 a	100.00 a	5.20 a	9.70 c	2.02 e
2.69	0	0	100.00 a	100.00 a	2.50 cd	14.83 ab	8.28 a
2.69	2.22	0	100.00 a	100.00 a	2.20 cd	14.30 ab	8.00 a
2.69	4.44	0	100.00 a	100.00 a	2.00 d	14.35 ab	7.91 a
2.69	8.88	0	100.00 a	100.00 a	3.00 bc	16.32 a	8.15 a
2.69	0	2.27	100.00 a	100.00 a	3.25 bc	11.40 bc	6.97 abc
2.69	0	4.54	100.00 a	100.00 a	3.00 bc	11.48 bc	4.85 bcd
2.69	0	9.08	100.00 a	100.00 a	3.80 b	11.43 bc	3.38 de

Medians with the same letter for each variable are not statistically different (Wilcoxon, α = 0.05). NAA: 1-naphthaleneacetic acid, BA: N6-benzyladenine, TDZ: thidiazuron, SUR: survival, SFE: shoot-forming explants, SE: shoots per explant, SL: shoot length, NRS: number of roots per shoot.

**Table 3 plants-15-00088-t003:** Effect of BA, TDZ, and NAA on shoot growth of *Dendrobium phalaenopsis* Sa-Nook ‘Thailand Black’ (second phase).

NAA	BA	TDZ	SUR	SFE	SE	SL	NRS
(µM)			(%)	(%)		(mm)	
0	0	0	100.00 a	100.00 a	2.65 ghi	47.6 ab	7.1 ab
0	2.22	0	100.00 a	100.00 a	3.16 fgh	39.6 cde	5.3 cd
0	4.44	0	100.00 a	100.00 a	2.91 ghi	35.2 de	5.0 cde
0	8.88	0	100.00 a	100.00 a	2.53 hi	43.0 bc	5.1 cde
0	0	2.27	100.00 a	100.00 a	5.13 bc	40.4 cd	6.7 ab
0	0	4.54	100.00 a	100.00 a	5.46 b	48.9 a	5.3 cde
0	0	9.08	100.00 a	100.00 a	8.54 a	39.9 cd	4.5 de
2.69	0	0	100.00 a	100.00 a	3.20 fg	29.0 f	5.5 cd
2.69	2.22	0	100.00 a	100.00 a	2.62 ghi	37.6 cde	7.4 a
2.69	4.44	0	100.00 a	100.00 a	2.40 i	40.5 cd	6.0 bc
2.69	8.88	0	100.00 a	100.00 a	3.61 ef	38.9 cde	4.8 de
2.69	0	2.27	100.00 a	100.00 a	4.27 de	36.6 de	4.5 de
2.69	0	4.54	100.00 a	100.00 a	4.61 cd	34.2 ef	4.2 e
2.69	0	9.08	100.00 a	100.00 a	5.56 b	37.0 de	4.9 cde

Means with the same letter for each variable are not statistically different (Wilcoxon, α = 0.05). NAA: 1-naphthaleneacetic acid, BA: N6-benzyladenine, TDZ: thidiazuron, SUR: survival, SFE: shoot-forming explants, SE: shoots per explant, SL: shoot length, NRS: number of roots per shoot.

**Table 4 plants-15-00088-t004:** Effect of growth regulators (NAA, BA, and TDZ) present in the shoot induction media on survival, new shoots, number of roots, and number of leaves of *Dendrobium phalaenopsis* Sa-Nook ‘Thailand Black’ plants during acclimatization.

NAA	BA	TDZ	SUR	NS	NR			NL		
(µM)			(%)		0	90	RC (%)	0	90	RC (%)
0	0	0	52.0 e	0.30 bcd	7.1 ab	7.5 bcd	5.9	3.8 a	2.4 ab	−37.0
0	2.22	0	60.0 de	0.25 cd	5.3 cd	7.1 cd	32.8	3.1 a	1.6 c	−48.7
0	4.44	0	24.0 f	0.00 d	5.0 cde	7.4 bcd	48.0	3.6 a	1.6 c	−55.1
0	8.88	0	64.0 cde	0.39 bcd	5.1 cde	6.5 de	27.1	3.3 a	1.4 c	−57.3
0	0	2.27	68.0 bcde	0.29 cd	6.7 ab	7.2 cd	8.9	3.2 a	1.9 bc	−41.4
0	0	4.54	84.0 abcd	0.35 bcd	5.3 cde	6.3 de	19.5	3.0 a	1.6 bc	−47.0
0	0	9.08	96.0 a	0.91 a	4.5 de	5.8 e	29.2	3.4 a	1.9 bc	−43.8
2.69	0	0	84.0 abcd	0.81 abc	5.5 cd	8.8 a	60.9	3.3 a	1.4 c	−56.4
2.69	2.22	0	84.0 abcd	0.76 abc	7.4 a	8.6 ab	16.4	3.0 a	3.1 a	4.4
2.69	4.44	0	72.0 abcde	0.60 abc	6.0 bc	8.0 abc	34.5	3.2 a	1.8 bc	−45.1
2.69	8.88	0	80.0 abcd	0.55 abcd	4.8 de	6.8 cde	42.8	3.0 a	1.9 bc	−37.0
2.69	0	2.27	92.0 ab	0.59 abcd	4.5 de	6.3 de	40.1	3.4 a	1.8 bc	−45.2
2.69	0	4.54	88.0 abc	0.51 abcd	4.2 e	5.7 e	37.8	3.2 a	1.8 bc	−43.0
2.69	0	9.08	88.0 abc	0.88 ab	4.9 cde	5.8 e	18.9	3.4 a	1.7 bc	−49.4

Means with the same letter in each column are not statistically different (Tukey, *p* < 0.05). NAA: 1-naphthaleneacetic acid, BA: N6-benzyladenine, TDZ: thidiazuron. RC: rate of change: increase (+) or decrease (−) concerning day zero. 0: start of acclimatization, 90: end of evaluation, SUR: survival, NS: new shoots, NR: number of roots, NL: number of leaves.

**Table 5 plants-15-00088-t005:** Effect of growth regulators (NAA, BA, and TDZ) present in the shoot induction media on root length, leaf length, and shoot length of *Dendrobium phalaenopsis* Sa-Nook ‘Thailand Black’ plants during acclimatization.

NAA	BA	TDZ	RL (mm)			LL (mm)			SL (mm)		
(µM)			0	90	RC (%)	0	90	RC (%)	0	90	RC (%)
0	0	0	42.3 ab	42.7 a	0.8	55.6 a	53.5 a	−3.8	47.6 ab	49.6 a	4.1
0	2.22	0	37.3 bcd	23.9 fg	−36.0	45.6 b	37.2 de	−18.4	39.6 cde	46.7 ab	17.9
0	4.44	0	42.9 a	22.1 g	−48.5	41.7 bc	28.2 f	−32.4	35.2 de	43.6 bc	23.9
0	8.88	0	39.6 abc	30.8 cd	−22.3	41.7 bc	35.1 e	−15.8	43.0 bc	46.6 ab	8.4
0	0	2.27	31.6 ef	25.3 fg	−19.9	42.0 bc	42.6 bcd	1.6	40.4 cd	41.1 cd	1.8
0	0	4.54	30.6 f	26.2 efg	−14.5	45.9 b	45.9 b	0.1	48.9 a	49.8 a	1.9
0	0	9.08	31.4 ef	26.6 def	−15.6	45.5 b	44.5 bc	−2.2	39.9 cd	44.0 bc	10.2
2.69	0	0	33.2 def	22.6 fg	−31.8	31.8 d	26.9 f	−15.4	29.0 f	31.3 f	7.7
2.69	2.22	0	30.9 f	30.4 cde	−1.5	41.6 bc	37.1 de	−11.0	37.6 cde	38.7 cde	3.1
2.69	4.44	0	31.1 f	24.4 fg	−21.6	41.0 bc	38.3 cde	−6.4	40.5 cd	40.8 cd	0.8
2.69	8.88	0	36.3 cde	31.7 bc	−12.5	44.3 bc	39.9 bcde	−9.9	38.9 cde	39.7 cde	2.1
2.69	0	2.27	35.2 cdef	33.0 bc	−6.1	39.5 c	35.7 e	−9.8	36.6 de	37.4 de	2.4
2.69	0	4.54	36.3 cde	35.8 b	−1.3	42.7 bc	38.3 cde	−10.4	34.2 ef	34.4 ef	0.6
2.69	0	9.08	33.9 def	30.4 cde	−10.2	46.4 b	41.6 bcde	−10.3	37.0 de	39.5 cde	6.7

Means with the same letter in each column are not statistically different (Tukey, *p* < 0.05). NAA: 1-naphthaleneacetic acid, BA: N6-benzyladenine, TDZ: thidiazuron. RC: rate of change: increase (+) or decrease (*−*) concerning day zero. 0: start of acclimatization, 90: end of evaluation, RL: root length, LL: leaf length, SL: shoot length.

**Table 6 plants-15-00088-t006:** Effect of growth regulators (BA, TDZ, NAA) on the production cost of acclimatized *Dendrobium phalaenopsis* Sa-Nook ‘Thailand Black’ plants.

NAA	BA	TDZ	Direct Costs	Indirect Costs	DM	Total Cost	NPL	Cost per Plant
(µM)			(USD)	(USD)	(USD)	(USD)		(USD)
0	0	0	275.23	142.10	23.81	441.14	198.94	2.22 b
0	2.22	0	275.26	142.10	23.81	441.17	240.80	1.83 bc
0	4.44	0	275.29	142.10	23.81	441.20	91.56	4.82 a
0	8.88	0	275.35	142.10	23.81	441.26	231.98	1.90 bc
0	0	2.27	276.16	142.10	23.81	442.07	391.72	1.13 def
0	0	4.54	277.09	142.10	23.81	443.00	483.84	0.92 ef
0	0	9.08	278.94	142.10	23.81	444.85	906.85	0.49 g
2.69	0	0	275.24	142.10	23.81	441.15	352.80	1.25 de
2.69	2.22	0	275.27	142.10	23.81	441.18	297.92	1.48 cd
2.69	4.44	0	275.30	142.10	23.81	441.21	246.33	1.79 bc
2.69	8.88	0	275.36	142.10	23.81	441.27	397.60	1.11 def
2.69	0	2.27	276.17	142.10	23.81	442.08	516.95	0.86 ef
2.69	0	4.54	277.10	142.10	23.81	443.01	490.00	0.90 ef
2.69	0	9.08	278.95	142.10	23.81	444.86	548.80	0.81 f

Means with the same letter are not statistically different (Tukey, *p* < 0.05). NAA: 1-naphthaleneacetic acid, BA: N6-benzyladenine, TDZ: thidiazuron. DM: depreciation and maintenance expenses, NPL: number of plants regenerated using one liter of shoot induction culture medium, and acclimatized. Cost calculations were conducted in U.S. dollars (USD) using the exchange rate at the close of trading on 21 August 2025, based on data from the Bank of Mexico, which was 18.77 MXN.

## Data Availability

The original contributions presented in this study are included in the article. Further inquiries can be directed to the corresponding author.

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
