# Peer review of "Effect of Culture Medium Composition on In Vitro Regeneration, Acclimatization, and Production Cost of Dendrobium phalaenopsis Sa-Nook ‘Thailand Black’ Plants"

_plants, 2025, doi:10.3390/plants15010088_

Round 1

Reviewer 1 Report

Comments and Suggestions for Authors

Generally, the manuscript should follow the regular structure: Introduction, Material and methods, Results and Discussion.

Introduction

WHat do you mean by illegal extraction? illegial collection from wild populations? Extraction is not the right word here

Instead of „This genius is distributed” use This species is common in tropical….

Mentioning cultivarnames like ‘Thailand Black’ or ‘Emma White’ or ‘Yuki White’ should always be Be enclosed in single quotation marks. Sometimes you write it correctly but most of the times not.

In Dendrobium phalaenopsis there are a few studies regarding its micropropagation. – this is not a good sentence. Instead: There are only a few studies regarding the micropropagation Dendrobium phalaenopsis.

vitroplant is not an English word use instead. in vitro plantlet, micropropagated plant, plant produced in vitro.

2.2 Acclimation should be acclimatization!

In the caption of Fig.2 NAA is mistyped like ANA

Discussion

failed to induce shoot formation in a significanly higher number OF EXPLANTS , please add this last two words otherwise it doesn’t make any sense.

Instead of [18] obtained you have to write Liu et al (2023) obtained………. [18].

Same with [25] observed, [31] indicate, reports by [38, 41, 42], [42] reported.

Discussion 2nd paragraph: Unlike the observed instead use Unlike what was observed in other Dendrobium species…

….since with 9.08 µM TDZ the greates number of shoots WERE OBTAINED.

Discussion 6th paragraph last sentence: ….and were beginning a new stage of growth. instead: and had begun a new stage of growth

Discussion 8th paragraph: PGR regulators – the word regulator is included in the PGR abbreviation, so use instead the PGRs applied

Discussion 9th paragraph: This knowledge helps TO determine

Material and methods?

Why did you use such unsusla and not uniform concentrations of the different hormons??? In this way it is rather hard to compare the effects.

Author Response

Comments 1: Generally, the manuscript should follow the regular structure: Introduction, Material and methods, Results and Discussion

Response 1: The manuscript structure was adjusted to align with the journal's guidelines, which require that the materials and methods section comes after the discussion.

Comments 2: What do you mean by illegal extraction? illegial collection from wild populations? Extraction is not the right word here

Response 2: Agree. The phase “illegal extraction” has been changed to: “illegal collection from wild population”  Page 1, line 30

Comments 3: Change: This genus is distributed to: This species is common

Response 3: Agree. The phrase “This genius is distributed” has been changed to: This species is common. Page 1, line 35

Comments 4: Mentioning cultivar names like ‘Thailand Black’ or ‘Emma White’ or ‘Yuki White’ should always be Be enclosed in single quotation marks. Sometimes you write it correctly but most of the times not.

Response 4: Agree. The names of the varieties were enclosed in quotation marks.  Page 2, line 44; Page 9, line 232; Page 9, line 238

Comments 5: In Dendrobium phalaenopsis there are a few studies regarding its micropropagation. – this is not a good sentence. Instead: There are only a few studies regarding the micropropagation Dendrobium phalaenopsis.

Response 5: Agree. The statement : “In Dendrobium phalaenopsis there are a few studies regarding its micropropagation” has been changed to: There are only a few studies regarding the micropropagation Dendrobium phalaenopsis. Page 2, line 51

Comments 6: Vitroplant is not an English word use instead. in vitro plantlet, micropropagated plant, plant produced in vitro.

Response 6: Agree. The term “vitroplant” has been replaced with: in vitro plantlet, micropropagated plant, or plant produced in vitro.  Page 2, line 69; Page 10, line 312; Page 10, line 322; Page 10, line 324; Page 10, line 327; Page 12, line 383

Comments 7: 2.2 Acclimation should be acclimatization

Response 7: Agree. The term “acclimation” has been changed to: acclimatization. Page 5, line 142

Comments 8: In the caption of Fig.2 NAA is mistyped like ANA.

Response 8: Agree. The word “ANA” has been changed to: NAA. Page 8, line 214

Comments 9: Failed to induce shoot formation in a significanly higher number OF EXPLANTS , please add this last two words otherwise it doesn’t make any sense.

Response 9:  This statement was deleted

Comments 10: Instead of [18] obtained you have to write Liu et al (2023) obtained………. [18]. Same with [25] observed, [31] indicate, reports by [38, 41, 42], [42] reported.

Response 10: The wording was changed to: Liu et al. (2023) obtained the highest number of shoots (6.1) from D. moniliforme with 4.44 μM BA + 5.37 μM NAA after 14 weeks of culture [18]. Subrahmanyeswari et al. (2022) observed the highest number of shoots per explant (12.3) when D. var. ‘Yuki White’ explants were cultured in the presence of 2.22 μM BA + 0.54 μM NAA + 217.2 μM adenine sulfate for eight weeks [25].  Page 9, lines 235-239; Page 9, line 237; Page 9, line 269; Page 10, line 320; Page 10, line 326.

Comments 11: Díaz et al. (2010) indicate that for orchid in vitro plantlet to ensure their survival after being transferred to ex vitro conditions, it is essential that they develop a sufficient number of leaves and a strong root system during the in vitro growth phase [31].

Response 11: Agree. The paragraph “[31] indicate that for orchid vitroplants to ensure their survival after being transferred to ex vitro conditions, it is essential that they develop a sufficient number of leaves and a strong root system during the in vitro growth phase” was changed to: Díaz et al. (2010) indicate that for orchid in vitro plantlet to ensure their survival after being transferred to ex vitro conditions, it is essential that they develop a sufficient number of leaves and a strong root system during the in vitro growth phase [31]. Page 9, lines 269-272.

Comments 12: Discussion 2nd paragraph: Unlike the observed instead use Unlike what was observed in other Dendrobium species…

Response 12: Agree. The phase “unlike the observed” has been changed to: “unlike what was observed in other Dendrobium species…” Page 9, line 240.

Comments 13: ….since with 9.08 µM TDZ the greates number of shoots WERE OBTAINED.

Response 13: This statement was deleted

Comments 14: Discussion 6th paragraph last sentence: ….and were beginning a new stage of growth. instead: and had begun a new stage of growth.

Response 14: Agree. The phase “and were beginning a new stage of growth” has been changed to: “and had begun a new stage of growth”  Page 10, lines 287-288.

Comments 15: Discussion 8th paragraph: PGR regulators – the word regulator is included in the PGR abbreviation, so use instead the PGRs applied.

Response 15: Agree. The phase “indicated that the PGR regulators applied during the regeneration phase..” was changed to: “indicated that the PGR applied during the regeneration phase”.  Page 10, lines 306-307

Comments 16: Discussion 9th paragraph: This knowledge helps TO determine

Response 16: Agree. The word “to” has been incorporated. Page 10, line 314.

Comments 17: Why did you use such unsusla and not uniform concentrations of the different hormons??? In this way it is rather hard to compare the effects.

Response 17: The concentrations of the growth regulators that were tested are uniform with respect to milligrams per liter.

Reviewer 2 Report

Comments and Suggestions for Authors

This work could be a valuable contribution to in vitro research on this species. However, for this to be the case, the authors should clarify the following methodological issues.

In Chapter 4.4, the authors distinguish two phases of shoot growth. The first lasts 90 days, after which the shoots are individually transferred to larger containers for the second growth phase. Why, then, do the authors not evaluate the effectiveness of the culture after the first phase (using the same parameters as for the second growth phase)?

The authors further state that after 120 days, they assessed the second growth phase of the culture, and the results are presented in Table 1. It is therefore expected that these plants will undergo an acclimatization process, but the authors do not explain in detail how this was accomplished. After the culture evaluation, were the shoots returned to the medium for an initial 5-day acclimatization period, or did the second growth phase last 120 days plus 5 days? In the first case, for example, the number of roots per shoot in Table 1 and Table 2 (column "acclimation start day 0") should be identical. In the second, how can we explain the change in their number over a period of only 5 days?

So what protocol should be used during micropropagation and acclimation?

Does shoot length in Table 1 and plant length in Table 2 mean the same thing? If not, define the difference.

Furthermore, the authors should indicate why the growth medium was supplemented with activated carbon during the second phase of shoot growth?

Author Response

Comments 1: In Chapter 4.4, the authors distinguish two phases of shoot growth. The first lasts 90 days, after which the shoots are individually transferred to larger containers for the second growth phase. Why, then, do the authors not evaluate the effectiveness of the culture after the first phase (using the same parameters as for the second growth phase)?

Response 1: Agree. The effectiveness of the culture after the first phase of shoot growth was shown in Table 2. Page 4, lines 123-127

Comments 2: The authors further state that after 120 days, they assessed the second growth phase of the culture, and the results are presented in Table 1. It is therefore expected that these plants will undergo an acclimatization process, but the authors do not explain in detail how this was accomplished. After the culture evaluation, were the shoots returned to the medium for an initial 5-day acclimatization period, or did the second growth phase last 120 days plus 5 days? In the first case, for example, the number of roots per shoot in Table 1 and Table 2 (column "acclimation start day 0") should be identical. In the second, how can we explain the change in their number over a period of only 5 days?

Response 2: Agree. The results from the second phase of shoot growth are presented in Table 3. Page 6, line 168-172

After completing the second shoot growth phase, the rooted shoots, still in the culture medium, were moved to an acclimatization tunnel in the greenhouse, where they stayed for 5 days for pre-acclimatization. Following this period, the plants were taken out of the jars and potted in the substrate. Page 11, lines 331-338

The number of roots per shoot in Table 3 and Table 4 (column day 0") are identical. Page 6, line 168-180

Comments 3: So what protocol should be used during micropropagation and acclimation?

Response 3: Agree. The protocol for micropropagation and acclimatization is described in the last paragraph of the discussion. Page 11, lines 331-338.

Comments 4: Does shoot length in Table 1 and plant length in Table 2 mean the same thing? If not, define the difference.

Response 4: Agree. The term “shoot length” in Tables 2 and 3 means the same thing that plant length (PL). In Table 5 PL has been changed to SL. Page 4, line 123; Page 6, line 168; Page 7, line 192

Comments 5: Furthermore, the authors should indicate why the growth medium was supplemented with activated carbon during the second phase of shoot growth?

Response 5: Agree. We explain the reason for supplementing the growth medium with activated charcoal during the second phase of shoot growth. Activated charcoal was utilized to absorb the phenols and ethylene released by the shoots, which promoting better development of both the aerial and root systems.  Page 9, lines 267-269.

Reviewer 3 Report

Comments and Suggestions for Authors

he manuscript is quite innovative since the study has been performed on Dendrobium phalaenopsis, Sa-Nook 'Thailand Black' variety which was not previously investigated for its response to micropropagation and TDZ application. The introduction is enough detailed and includes the most important aspects regarding the importance of this species as ornamental plants. The previous studies on tissue culture studies for propagation of this species were also included putting in evidence the relevant points suggesting the necessity of using micropropagation in this species directing to the importance of the approach of the study performed.

The material and method are well described

What I have to underline is that it seems that all the results of morphogenic response concerning the application of different growth regulators presented concern the data collection performed after 30 days of treatment with growth regulators (GR)  plus other 120 days of culture performed on basal medium with 1 g/L of activated carbon without GR (second phase of shoot growth), as reported in the material and methods. For this reason, I think that the effect of the growth regulators could has been highly mitigated. For this reason, I strongly suggest collecting morphological results after the first 30 days of culture on the growth regulators including in the paper the data collected and figures. In addition, performing further studies keeping the explants for more subcultures in this condition could be also useful to better understand the role of the growth regulators tested, giving also more information on their morphological effect.

Description of the results has, on my opinion, other some critical points that should be taken into consideration.

In fact, I think, the report of data collected should be, in some cases, reconsidered checking the results of the statistical analysis. Looking at the description of data reported in Table 1, the below reported lines in the text

…. the lowest values for both variables were observed with 4.44 µM BA + 2.69 µM NAA (Table 1).

Shoots regenerated with 2.22 µM BA (Figure 1E) were 12 % taller (17.9 mm) than the control (Table 1, Figure 1A). The highest number of roots per shoot was recorded with any of the BA concentrations tested individually or in combination with NAA, without these values being significantly different from the control. Shoots obtained with TDZ formed the lowest number of roots, except when 2.27 µM TDZ was combined with 2.69 µM NAA.

Concerning Table 2, I strongly suggest dividing it in 2 tables making easier the description and the interpretation.

Also, In the description in the text of data included in table 2 there are, on my opinion, some revisions to be done according to the results of the statistical analysis and taking better into consideration that similar lowercase letters indicating not significant differences among treatments

Concerning paragraph 2.3 The production costs, I do not think that is appropriate including this type of data in a scientific paper.

Taking into consideration all the above points, I have to suggest a large revision of the manuscript including more studies to make it suitable for publication on this journal.

Author Response

Comments 1: Concerning Table 2, I strongly suggest dividing it in 2 tables making easier the description and the interpretation

Response 1: Agree. Table 2 has been divided into two separate tables, now referred to as Table 4 and Table 5. Page 6, lines 174-180; Page 7, line 191-197.

Comments 2: What I have to underline is that it seems that all the results of morphogenic response concerning the application of different growth regulators presented concern the data collection performed after 30 days of treatment with growth regulators (GR)  plus other 120 days of culture performed on basal medium with 1 g/L of activated carbon without GR (second phase of shoot growth), as reported in the material and methods. For this reason, I think that the effect of the growth regulators could has been highly mitigated. For this reason, I strongly suggest collecting morphological results after the first 30 days of culture on the growth regulators including in the paper the data collected and figures..

Response 2: Agree.

Table 1 shows the results obtained from the 30 days of culture (induction phase).

It was found that during the induction phase, the number of regenerated shoots depended on the type of PGR and its concentration in the culture medium. Just as the number of shoots obtained during the induction phase depended on the type and concentration of PGR, the number of shoots, shoot length, and number of roots during the first and second shoot growth phases were affected by the type and concentration of the PGR tested during induction, even though these regulators were no longer present in the shoot growth medium. Page 3, lines 92-96; Page 9, lines 248-252.

Comments 3: In fact, I think, the report of data collected should be, in some cases, reconsidered checking the results of the statistical analysis. Looking at the description of data reported in Table 1, the below reported lines in the text …. the lowest values for both variables were observed with 4.44 µM BA + 2.69 µM NAA (Table 1). Shoots regenerated with 2.22 µM BA (Figure 1E) were 12 % taller (17.9 mm) than the control (Table 1, Figure 1A). The highest number of roots per shoot was recorded with any of the BA concentrations tested individually or in combination with NAA, without these values being significantly different from the control. Shoots obtained with TDZ formed the lowest number of roots, except when 2.27 µM TDZ was combined with 2.69 µM NAA. 

Response 3: Agree. The description of the results of the statistical analysis shown in Table 1 (now Table 2 and Table 3) has been reviewed.  Page 3, lines 98-112; Page 4, line 128-138.

Comments 4: Concerning paragraph 2.3 The production costs, I do not think that is appropriate including this type of data in a scientific paper.

Response 4: We believe that plants produced through micropropagation are often sold. To evaluate the effectiveness of the developed protocol, it is essential to determine if the costs of these plants are competitive with those propagated by other methods.  

Round 2

Reviewer 3 Report

Comments and Suggestions for Authors

The article has been accurately improved. Concerning the insertion of production costs, I have still some doubts that is appropriate including this type of data in a scientific paper, but I think that, considering the response of the authors, it is  up to the editor making the final decision.